# Accelerating Global Sensitivity Analysis via Supervised Machine Learning Tools: Case Studies for Mineral Processing Models

**Freddy A. Lucay** 

Escuela de Ingeniería Química, Pontificia Universidad Católica de Valparaíso, Valparaíso 2340000, Chile; freddy.lucay@pucv.cl

**Abstract:** Global sensitivity analysis (GSA) is a fundamental tool for identifying input variables that determine the behavior of the mathematical models under uncertainty. Among the methods proposed to perform GSA, those based on the Sobol method are highlighted because of their versatility and robustness; however, applications using complex models are impractical owing to their significant processing time. This research proposes a methodology to accelerate GSA via surrogate models based on the modern design of experiments and supervised machine learning (SML) tools. Three case studies based on an SAG mill and cell bank are presented to illustrate the applicability of the proposed procedure. The first two consider batch training for SML tools included in the Python and R programming languages, and the third considers online sequential (OS) training for an extreme learning machine (ELM). The results reveal significant computational gains from the methodology proposed. In addition, GSA enables the quantification of the impact of critical input variables on metallurgical process performance, such as ore hardness, ore size, and superficial air velocity, which has only been reported in the literature from an experimental standpoint. Finally, GSA-OS-ELM opens the door to estimating online sensitivity indices for the equipment used in mineral processing.

**Keywords:** global sensitivity analysis; supervised machine learning; mineral processing



## 1. Introduction

The mining industry has been the economic engine in various countries worldwide, such as Chile, Perú, and South Africa. In the case of Chile, mining projects face challenges such as a reducing deposit grade, water stress, energy consumption, and environmental impact (mainly generating tailings), configuring an adverse scenario to maintain the current production level. In addition, there is a mineralogical change in the deposits, specifically, a change from copper oxide ores to copper sulfide ores, that is processed by froth flotation. This requires a liberated ore processed as pulp, thus increasing energy and water consumption. According to Montes [1], it has been estimated that minerals treated by froth flotation will increase from 574 million tons in 2017 to 1004 million tons in 2029. Under this scenario, the operating conditions of the processes implemented in the sulfide ore value chain must be optimized to mitigate the problems mentioned earlier.

Mineral processing can be optimized using simulation systems such as ModSim, MetSim, MolyCop-Tools, or JkSIMmet. These systems are powerful tools and have demonstrated their usefulness in many process applications [2–7]. However, such optimizations implement fixed operating parameters, unlike industrial practice, because mining processing equipment operates under uncertainty that affects its metallurgical performance. For instance, the hardness of the ore processed in comminution systems exhibits geological uncertainty, i.e., it presents a variation inherent to mining deposits whose increment implies an increase in the energy required to liberate the valuable mineral. In this respect, the ore size exhibits operating uncertainty in flotation circuits, i.e., there is a lack of knowledge related to the insufficient measurements or assumptions made in the plant. In both instances,

the impact of uncertainty could generate poor metallurgical performances; therefore, this must be quantified to efficiently optimize the process studied.

The effect of the uncertainty on mathematical models, the basis of simulation systems, can be addressed via global sensitivity analysis (GSA). GSA can be applied using different approaches, such as screening, linear regression-based, and variance decomposition-based methods, the latter highlighted for their efficiency and versatility. Within this context, variance decomposition-based methods have been applied in mineral processing, e.g., flotation [8], grinding [9], mineral leaching [10], and lithium ore processing [11]. However, these works did not quantify the impact of critical input variables on the metallurgy performance of the processes; for example, the effect of ore hardness on grinding power consumption or the impact of superficial air velocity and particle size on flotation copper grade was not studied. Another aspect to consider is that GSA requires intensive computations [12], making it unfeasible to study the effect of uncertainty using simulation systems or robust mathematical models. As a result, surrogate models (SMs) have arisen as a feasible alternative for avoiding this drawback.

SMs are an engineering tool used when an outcome of interest cannot easily be directly measured, so an outcome model is used instead. In other words, the primary motivation to develop surrogate modeling strategies is to make better use of the available, usually limited, computational burden [13]. SMs are also known as response surface models, metamodels, emulators, or proxy models [14], and they have been implemented in mineral processing [15,16]. However, these works exhibit the following disadvantages: (a) they do not compare different approaches; (b) they do not consider the uncertainty; (c) they do not compare libraries/packages; (d) they consider only batch learning. The framework implemented to construct surrogate models includes three components: first, an experiment design guides the computational experiments; second, an algorithm repeatedly runs the original computational model in the experiments proposed by the DoE and collects their responses; third, the collected information is used to train or fit a mathematical technique. Among the mathematical methods used to construct surrogate models is highlighted machine learning.

Machine learning (ML) is a branch of computational intelligence having an ever-increasing presence and impact on a wide variety of research and commercial fields. ML includes tools focus on endowing programs with the ability to learn and adapt, and they are classified according to the training type: supervised, non-supervised, and reinforcement [17]. The first requires labeled data samples to approximate mapping to predict output values or data labels; here, we find regression and classification problems. The second does not demand labeled data, and its goal is to model the underlying structure or distribution in the data; here, we find clustering and dimensionality reduction problems. The third is a robust mathematical framework for experience-driven autonomous learning. There are multiple libraries/packages for supervised machine learning included in different programming languages, for example, Python, R, and Julia [17–19].

This work presents a methodology to construct SM for complex models related to devices implemented in mineral processing, aiming to accelerate global sensitivity analysis. The procedure includes DoE and SML tools and the Sobol–Jansen method to perform the GSA. To determine the best tools to build SMs, benchmarking is confected considering libraries/packages included in the Python and R programming languages. These libraries/packages offer SML tools trained via exact algorithms, and to enhance this benchmarking, the technique known as extreme learning machine (ELM) was coded by the author and trained via an approximate algorithm. Three case studies based on an SAG mill and a flotation cell bank are presented to demonstrate the applicability of the methodology proposed.

## 2. Methods

This section presents a brief review of the methods supporting the methodology proposed to accelerate the global sensitivity analysis for metallurgical models.

### 2.1. Global Sensitivity Analysis (GSA)

The sensitivity analysis (SA) helps to detect the contribution of the uncertainties of the input variables to the uncertainties of the output variables [20]. SA can be divided into local or global: the first is based on the local derivatives of the output variable concerning one input variable; the second considers evaluating the output variable variability considering the simultaneous fluctuation of input variables in their uncertainty domains [12]. GSA can be performed using different approaches, with an emphasis on those based on variance decomposition because of their efficiency and versatility. The Sobol–Jansen method belongs to this category and allows the determination of the first-order sensitivity index and the total sensitivity index for each input variable of the model [21]. The first-order index establishes the most significant input variable on the output variable. The total index provides information to identify the input variables that do not affect the output variable. According to the literature [22], the Sobol–Jansen method exhibits good performance in analyzing processes utilized in mining. However, this method consumes a significantly large processing time when the model analyzed is complex.

### 2.2. Design of Experiments (DoE)

Design of Experiments is a branch of statistics that helps designers plan, execute, and analyze tests to evaluate the process responses. DoE employs different space-filling strategies to empirically capture the behavior of the underlying system over limited ranges of variables [23]. DoE can be divided into two families, which are classical and modern experiment design. The first is based on laboratory experiments; examples of this category are factorial design, central composite design, and Box–Behnken design. The second is based on computer simulations to replace the expensive physical experiments with faster and cheaper computer simulations; examples of this category include Latin hypercube sampling, symmetric sampling, and orthogonal array sampling [13].

### 2.3. Machine Learning (ML)

Machine learning can be defined as the science (and art) of programming computers so that they can learn from datasets. According to the literature, the first real ML application was the spam filter; afterwards, the spam filter was followed by hundreds of ML applications that now quietly power hundreds of products [17]. Behind these applications is the resolution of regression, classification, clustering, and dimensionality reduction problems, among others, the first of these being the spotlight of the current work. This problem considers a labeled dataset to approximate mapping to predict output values and can be addressed via tools included in SML. Among the latter can be found the support vector machine, multilayer perceptron, Gaussian process, and extreme learning machine, among others—the interested reader can find a review in [17,24]. These tools are available in scikit-learn [25] and TensorFlow libraries [26] for the Python programming language, and e1071 [27], RandomForest [28], Neuralnet [29], and Gaupro packages for the R programming language. In this work, they are compared to determine the best alternatives for developing SMs for minerals processing.

Table 1 shows a detailed breakdown of the programming language, library/package, and SML tools implemented to construct the SMs. Note that the author has coded ELM, which will be addressed in more detail in the following section, to introduce the procedure used for online sequential training.

Extreme Learning Machine (ELM)

ELM was proposed by Guang et al. [30] and it considers a single-layer feedforward neural network. Unlike most machine learning tools based on descent gradient, ELM assigns random values to the weights between the input and hidden layer, and biases in the hidden layer, and they are fixed during the training stage. The expert reader will have noticed that ELM is based on concepts related to RBF networks training; furthermore, some researchers have indicated that there is intelligent plagiarism (rewriting others' results) by

Guang and co-workers [31]. Aside from this controversy, ELM has been utilized successfully in many tasks, such as regression, clustering, and classification problems [32,33]. Below, we briefly describe ELM, assuming that the entire dataset is available during the training stage.

**Table 1.** Tools implemented to construct SMs.

| Supervised Machine Learning Tools | Python | R |
|---|---|---|
| Support vector regression (SVR) | Scikit-Learn | e1071 |
| Random Forest regression (RFR) | Scikit-Learn | RandomForest |
| Multilayer perceptron (MLP) | Scikit-Learn, Tensorflow | Neuralnet |
| Linear regression (LR) | Scikit-Learn | Stats |
| Elastic net regression (ENR) | Scikit-Learn | - |
| Ridge regression (RR) | Scikit-Learn | - |
| Lasso regression (LAR) | Scikit-Learn | - |
| Gaussian process regression (GPR) | Scikit-Learn | Gaupro |
| Extreme learning machine (ELM) | - | Authors |

We consider a training dataset $\aleph_0 = \left\{ (x_j, t_j) \right\}_{j=1}^{N_0}$ where $x_i \in \mathbb{R}^n$ is an input vector and $t_i \in \mathbb{R}^m$ is a target vector. The output of a single hidden layer feedforward neural network with $\widetilde{N}$ hidden nodes can be expressed as follows

$$f(x_j) = \sum_{i=1}^{\widetilde{N}} \beta_i h_i(x_j) = h(x_j)\beta, \ j = 1, 2, \ldots, N_0 \tag{1}$$

where $\beta = \left[ \beta_1, \ldots, \beta_{\widetilde{N}} \right]^T$ is the output weight vector between the hidden layer of $\widetilde{N}$ nodes and the $m \geq 1$ output nodes, and $h(x_j) = \left[ h_1(x_j), \ldots, h_{\widetilde{N}}(x_j) \right]^T$ is the output vector of the hidden layer concerning the inputs $x_j$. In real applications, $h_i(x_j)$ can be expressed as: $h_i(x_j) = G(a_i, b_i, x_j)$, where $a_i \in \mathbb{R}^n$ and $b_i \in \mathbb{R}$ are the learning parameters of hidden nodes, and $G$ is a nonlinear function satisfying ELM's universal approximation capability theorems. Some functions reported in the literature are sigmoid, hyperbolic tangent, Gaussian, multiquadric, hard limit, and cosine. The training procedure for ELM considers random initialization of $a_i$ and $b_i$, facilitating the definition of the matrix $H$ as follows:

$$H(\alpha_1, \ldots, \alpha_{\widetilde{N}}, b_1, \ldots, b_{\widetilde{N}}, x_1, \ldots, x_N) = \begin{bmatrix} G(\alpha_1, b_1, x_1) & \cdots & G(\alpha_{\widetilde{N}}, b_{\widetilde{N}}, x_{\widetilde{N}}) \\ \vdots & \ddots & \vdots \\ G(\alpha_1, b_1, x_N) & \vdots & G(\alpha_{\widetilde{N}}, b_{\widetilde{N}}, x_{\widetilde{N}}) \end{bmatrix}_{N \times \widetilde{N}}$$

Here, $H$ is called the hidden layer output matrix of the network. Afterwards, $\beta$ is determined by solving

$$\min_{\beta \in \mathbb{R}^{\widetilde{N} \times m}} \|H\beta - T\|^2 \tag{2}$$

where

$$\beta = \begin{bmatrix} \beta_1^T \\ \vdots \\ \beta_{\widetilde{N}}^T \end{bmatrix}_{\widetilde{N} \times m} \text{and } T = \begin{bmatrix} t_1^T \\ \vdots \\ t_N^T \end{bmatrix}_{N \times m} = \begin{bmatrix} t_{11} & \cdots & t_{1m}) \\ \vdots & \ddots & \vdots \\ t_{N1} & \vdots & t_{Nm}) \end{bmatrix}_{N \times m} .$$

Here, $T$ is the training data target matrix, and $\|\cdot\|$ is the Frobenius norm. The solution of Equation (2) is obtained by differentiating $\|H\beta - T\|^2$ with respect to $\beta$ and equalizing it to zero, obtaining $2H^T(H\beta - T) = 0$ or, equivalently, $\beta = H^\dagger T$, where $H^\dagger = (H^T H)^{-1} H^T$ is called the Moore–Penrose generalized inverse of matrix $H$. The reader interested in the detailed mathematical formulation and training procedure of ELM can see [33].

According to a related article [34], the optimization of random initialized weights and biases of ELM can be addressed using metaheuristic algorithms, classified as swarm

intelligence, physics-based, evolutionary, and human-based algorithms. Evolutionary algorithms (EAs) use mechanisms inspired by biological evolution for solving problems such as reproduction, mutation, recombination, and selection [35,36]. Some examples of EAs are genetic algorithms, evolutionary strategies, and differential evolution algorithms (DEAs) [35]. In this work, the latter was implemented to optimize the weights and biases of batch ELM and was labeled as ELM-DEA.

On the other hand, as we commented earlier, the batch ELM assumes that the entire dataset is available for training; however, in specific applications, sometimes we may not have access to the whole dataset because new samples are being added. Then, we need to retrain the ELM every time the dataset grows. However, the new samples often account for only a small part, so it is inefficient to repeatedly retrain the ELM using the whole dataset. Hence, the ELM's initial formulation must be modified. Liang et al. [32] proposed the online sequential ELM (OS-ELM), which can learn the dataset chunk-by-chunk, and it is summarized as follows [34].

---

**Algorithm 1.** The OS-ELM algorithm.

---

**Input:** $\aleph_0 = \left\{ \left( x_j, t_j \right) \right\}_{j=1}^{N_0}$

**Output:** A trained ELM model

*Initialization phase:*

**Set $t = 0$.**

Calculate $H_o$ using the training set and random parameters.

Obtain the output weight using $\beta_o = M_o H_o^T T_o$ where $M_o = \left( H_o^T H_o \right)^{-1}$

*Online sequential learning phase:*

For each new block of data, update the layer output matrix as $H_{t+1} = [h_t; h_{t+1}]$, where $h_{t+1}$ denotes the hidden output of the new data block.

Update the output weights as $\beta_{t+1} = \beta_t + M_{t+1} h_{t+1} \left( t_i^T - h_{t+1}^T \beta_t \right)$, where $M_{t+1} = M_t - M_t h_{t+1} h_{t+1}^T M_t \left( 1 + h_{t+1}^T M_t h_{t+1} \right)^{-1}$

**Set $t = t + 1$**

---

### 2.3.1. Methodology

Figure 1 shows the procedure proposed to construct SMs. The blue boxes represent the main structure of the methodology, which is described as follows:

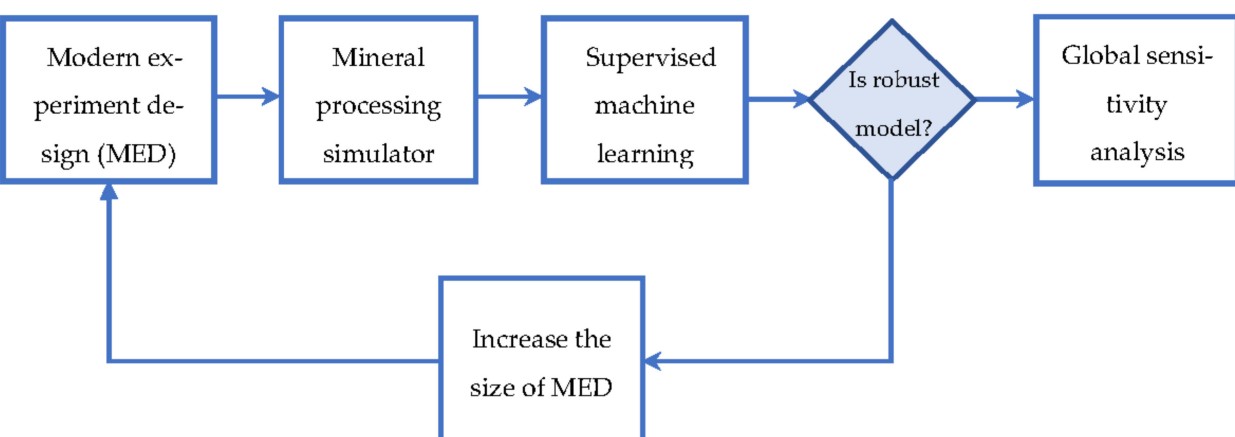

**Figure 1.** Flowchart for building SMs for accelerating GSA.

Step 1: The user must select a set of input variables for the process studied and their relevant operating ranges. Then, a modern experiment design is used to sample the space generated by input variables.

Step 2: An algorithm runs the mining processing model following the previous experiment design. The information generated is collected and processed to filter noise and outliers.

Step 3: The collected information is implemented to train an SML tool. The latter tends to overtrain; thus, the initial sample is divided into training and testing datasets. The first is used to construct the SM, and the second is utilized to validate it. The measure called the coefficient of determination (R2) is used to evaluate the statistical performance of the SML tool, which is considered robust if it exhibits an R2 greater than 0.80 in training and testing. If the SM is not robust, an increase in the size of the experimental design is considered.

Step 4: The GSA is carried out using a Sobol-based method and the SM.

It is worth mentioning that there are several measures to evaluate the performance of surrogate models. Prominent among these are R2, mean squared error (MSE), root mean squared error (RMSE), and mean absolute error (MAE) [37]. The latter three can vary from 0 to any larger number, whereas R2 exhibits values between 0 and 1. An R2 value closer to 1 means that the regression model covers most parts of the variance of the values of the response variable and can be termed as a good model. In contrast, with MSE, RMSE, and MAE values depending on the scale the of values of the response variable, the value will be different, and hence, it is difficult to assess for certain whether the regression model is good or otherwise.

## 3. Applications

This section presents three examples to illustrate the applicability of the methodology proposed. It is worth mentioning that all the experiments were carried out in JupyterLab using R and Python kernels and a computer with an Intel Core i7 2.21 GHz and 16 GB of RAM.

### 3.1. SAG Mill: Batch Training

We considered the SAG mill shown in Figure 2, modeled using mathematical expressions reported by [9] and a model to estimate the ore hardness ($\gamma$) in the feed. Specifically, $\gamma(t) = m_\gamma(\Gamma(t) - 1) + \gamma_{max}$, where $m_\gamma = (\gamma_{max} - \gamma_{min})/(\Gamma_{min} - \Gamma_{max})$; $\Gamma$ is the ore hardness in the Mohs scale, $\Gamma_{min}$ is minimum hardness in the Mohs scale, $\Gamma_{max}$ is the maximum hardness in the Mohs scale, $\gamma_{max}$ is equal to 1.5, and $\gamma_{min}$ is equal to 0.5. For more detail on the hardness model, see [38]. The grinding model was coded in the R programming language and solved via the nleqslv solver, assuming that the SAG mill behaved like a perfectly mixed reactor with first-order kinetics. The input variables considered in the model are shown in Table 2.

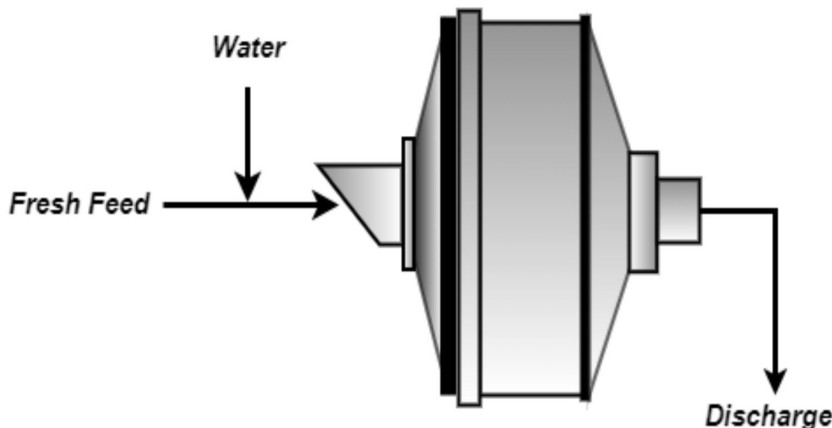

**Figure 2.** Schematic representation of an SAG mill.

**Table 2.** Input variables from the SAG mill model.

| Label | Input Variables | Operational Conditions | Unit | Uncertainty |
|---|---|---|---|---|
| $x_1$ | Feed Flowrate ($F$) | 3.45 | t/h | $U[3.34, 3.55]$, |
| $x_2$ | Steel ball volume occupation in the mill ($J_b$) | 8.5 | % | $U[6.37\%, 10.6\%]$ |
| $x_3$ | Discharge solid percentage ($X_d$) | 74 | % | $U[70.3\%, 77.7\%]$ |
| $x_4$ | Percentage of critical speed ($\phi_c$) | 72 | % | $U[70.56\%, 73.44\%]$ |
| $x_5$ | Size fraction $> 4''$ ($f_1$) | 11 | % | $N[11\%, 0.4]$ |
| $x_6$ | Size fraction $4''$–$2''$ ($f_2$) | 8.5 | % | $N[8.5\%, 0.6]$ |
| $x_7$ | Size fraction $< 2''$ ($f_3$) | 80.5 | % | $N[80.5\%, 0.87]$, |
| $x_8$ | Ore hardness ($\Gamma$) | 6 | Mohs scale | $N[6, 0.3]$ |

The power consumption ($P_w$) and comminution specific energy ($E_{cs} = P_w/F$) were selected as the model output variables because of their relevance to the total cost in the mining industry, representing values between 60 to 80% of the electric costs. The distribution functions used to describe the uncertainty of the input variables are shown in Table 2. The GSA was performed using the Sobol–Jansen function included in the sensitivity package for the R programming language. The Python programming language offers several libraries for carrying out GSA; however, they do not include the Sobol–Jansen method. Within this context, this method was coded by the author (see Supplementary Material). The Sobol–Jansen method requires: (a) a mathematical model; (b) two subsamples of the same size; (c) a resampling method for estimating the variance of sensitivity indices. According to Pianosi et al. [39], the sample size commonly used is 500 to 1000 [20]. However, the researcher indicated that the sample size may vary significantly from one application to another. Therefore, a much larger sample might be needed to achieve reliable results. Additionally, the number of samples required to achieve stable sensitivity indices can vary from one input variable to another, with low sensitivity inputs usually converging faster than high sensitivity ones [40]. In this first instance, a sample size equal to 1000 allows for the achievement of stable sensitivity indices. Therefore, the sample size used for carrying out GSA was 16,000 data. The bootstrapping technique consisting of random sampling with replacement from the original data was used as the resampling method. The number of resampling methods used was equal to 100, which is consistent with the literature.

The execution time of the Sobol–Jansen function applied to the grinding model was approximately 18,500 s, revealing that the complexity of the model and sample size severely affected the applicability of this method, and consequently, the analysis of the results. To reduce the execution time, the flowchart presented in Figure 1 was applied to construct SMs. In Step 1, the LHS sampling method was used to generate the operational conditions of the SAG mill. In Step 2, previous samples were used to simulate the grinding model. In Step 3, the samples, with their corresponding responses, were used to construct SMs, whose results are shown in Figure 3.

Figure 3 shows the results obtained during the SML tool training and testing using 80 and 20% of the dataset, respectively. Here, the tools labeled with py and r were constructed in Python and R, respectively. Note that SML tool multiparameters were tuned via trial and error. For instance, for MLP, this procedure evaluated the effect of the multiparameters on its statistical performance, including the number of neurons by layer, activation function, training rate, training algorithm, and the number of layers. Figure 3 reveals that as the dataset increased from 100 to 1000, and that R2 increased for both SML tool training and testing. This increase in samples helped to improve the capture of the SAG mill's behavior on uncertainty space, and consequently, the SML tool's performance. This figure shows that SVR-py, MLP-skl, LR-skl, ENR-py, RR-py, LAR-py, GPR-py, MLP-TensorFlow-py, MLP-r, SVR-r, LR-r, and ELM-DEA-r provided a better yield than RFR-py and RFR-r, but all exhibited an R2 greater than 0.8 when the simulation sample was equal to 1000; then, all SML tools moved to the next stage. Before they proceeded to the next step, note that RFR-r and RFR-py exhibited disparate performances. RFR, offered by the RandomForest package

(RFR-r) and the scikit-learn library (RFR-p), is based on Breiman's original version [41] and Geurts et al. [42], respectively. The latter consists of heavily randomizing both input variables and cut-point while splitting a tree node. In the extreme case, this approach randomly chooses a single input variable and cut-point at each node, building totally randomized trees whose structures are independent of the output variable values of the learning sample. Therefore, the RFR versions offered by scikit-learn and RandomForest differ, which could explain the disparity in their performance. In the case of RFR-r, its decreasing performance could be related to perturbations induced in the models by modifying the algorithm responsible for the search of the optimal split during tree growing [42]. In this context, the researcher indicated that it is productive to somewhat deteriorate the algorithm's "optimality" that seeks the best split locally at each tree node. In Step 4, the GSA was carried out using the Sobol–Jansen method and SMs, and their results are shown in Figures 4 and 5.

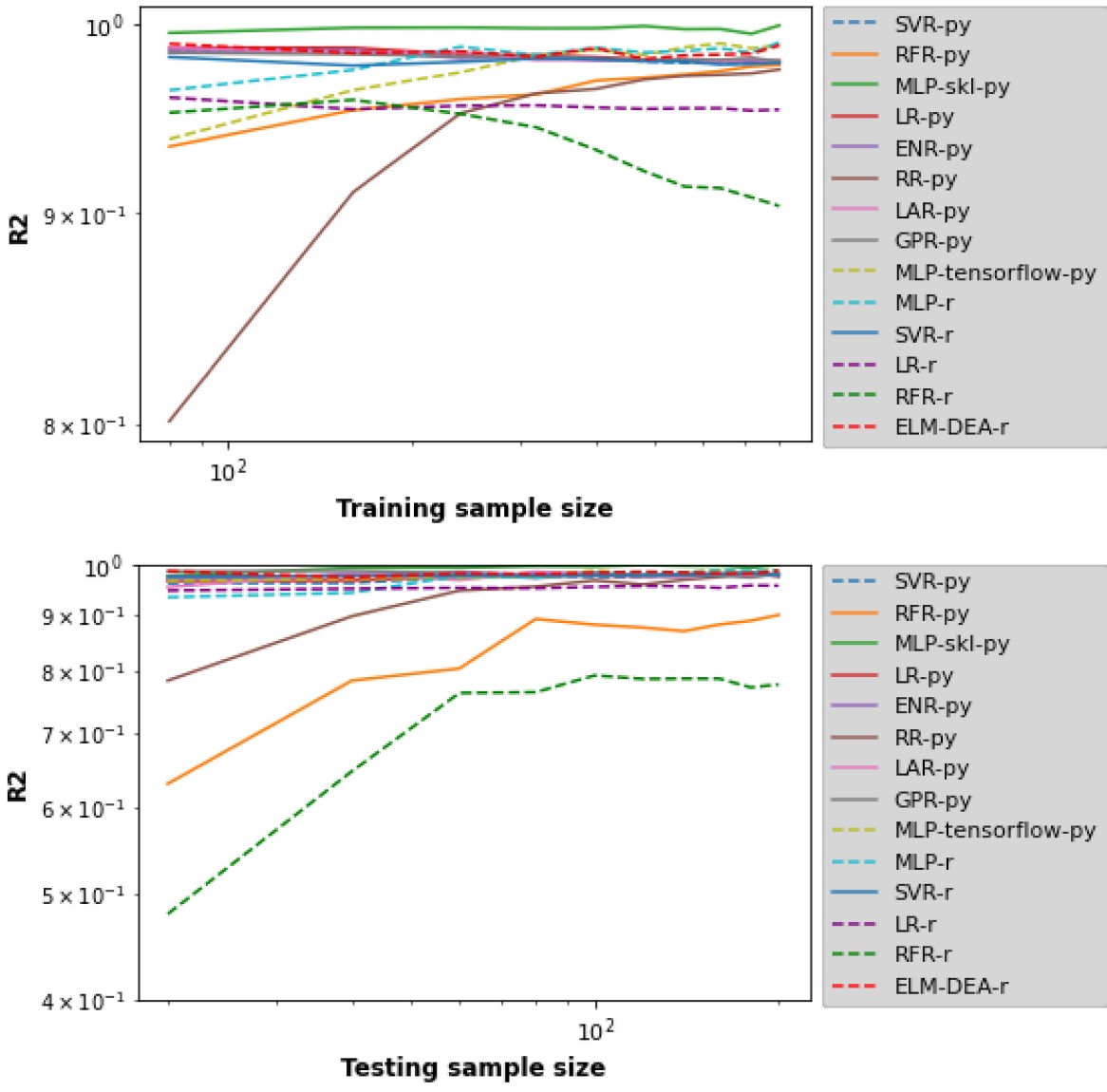

**Figure 3.** Performance of SML tools to construct SMs for grinding model outcomes.

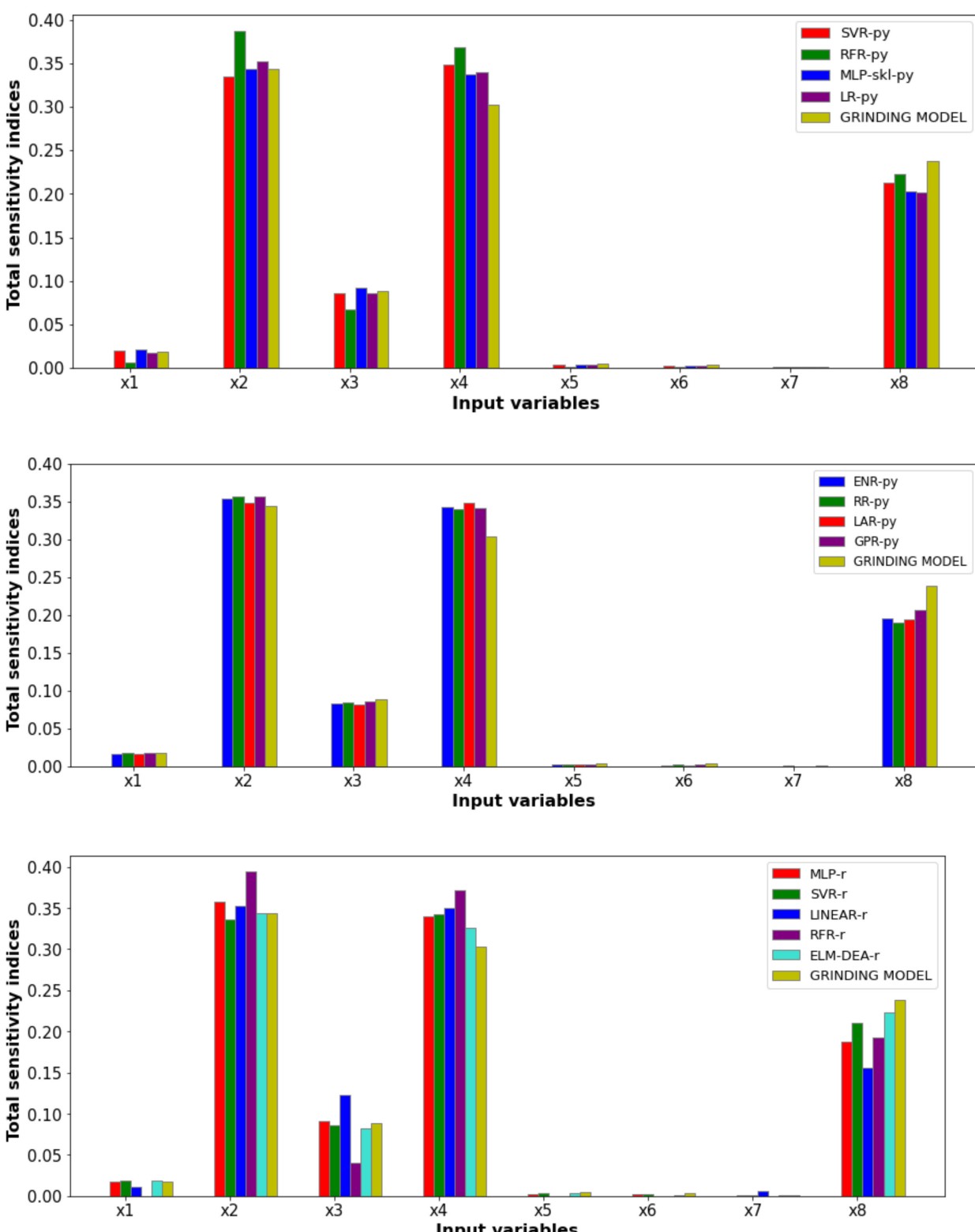

**Figure 4.** Total sensitivity indices obtained via Sobol–Jansen method and SMs for SAG mill power consumption.

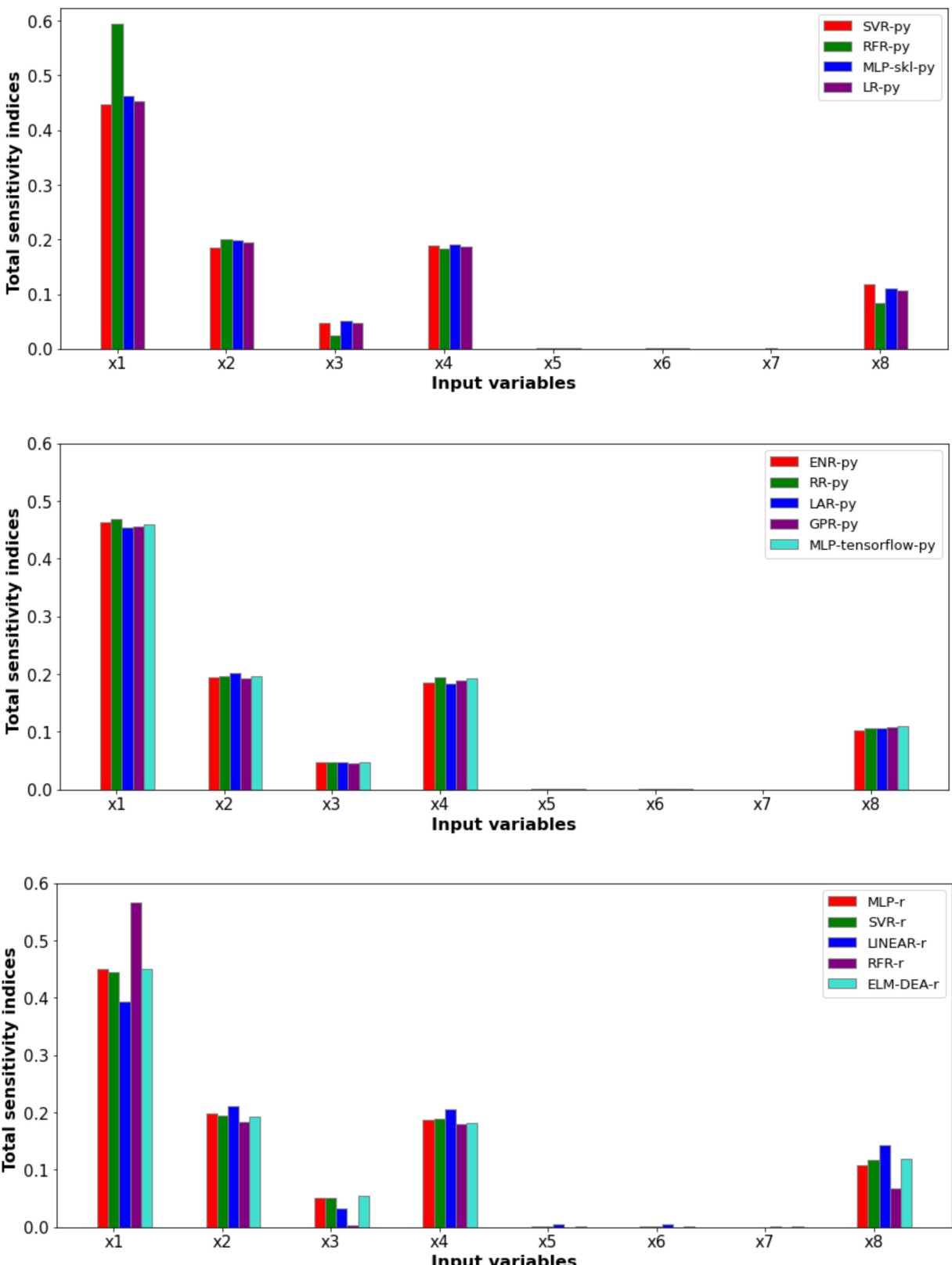

**Figure 5.** Total sensitivity indices obtained via the Sobol–Jansen method and surrogate models for comminution-specific energy.

Figure 4 shows the total sensitivity indices provided by the Sobol–Jansen method for the surrogate and grinding models when the output variable was the power consumption. Here, we can see that the total sensitivity indices obtained from SM were similar to those obtained with the grinding model, highlighting the indices reached with the ELM-DEA surrogate model, which required more training time (approximately 600 s) owing to the stochastic nature of DEA. The GSA indicated that the influential input variables on power consumption were $J_b$, $\phi_c$, $X_d$, and $\Gamma$. The equation used to estimate the power consumption included the first two input variables, explaining their high total Sobol index [43]. $X_d$ influenced the ratio between the ore mass and water mass retained inside the mill, and therefore, the power consumption. $\Gamma$ influenced the specific breakage rate, and consequently, the mass retained in the mill. The latter influenced a fraction of mill filling, and in turn, the power consumption (see [9,43]). Figure 5 shows the total sensitivity indices provided by the Sobol–Jansen method for the SMs when the output variable was the comminution-specific energy. The latter was influenced mainly by $F$, $J_b$, $\phi_c$, and $\Gamma$. By definition, $F$ directly affects the comminution-specific energy, explaining this input variable's high total Sobol index. $J_b$ and $\phi_c$, and $\Gamma$, directly and indirectly, affected the power consumption, respectively, and therefore the comminution-specific energy [9,38,43]. An immediate result of the GSA was reducing the uncertainty space to ($F = 3.45$, $J_b$, $X_d$, $\phi_c$, $f_1 = 11$, $f_2 = 8.5$, $f_3 = 80.5$, $\Gamma$) and ($F$, $J_b$, $X_d = 74$, $\phi_c$, $f_1 = 11$, $f_2 = 8.5$, $f_3 = 80.5$, $\Gamma$) for the power consumption and comminution-specific energy, respectively. Therefore, optimizing the SAG mill's outcomes must be made over the reduced spaces to achieve effective operational conditions and decrease the computational burden. Such optimization can be addressed via metaheuristic algorithms [44]. Table 3 shows the execution time required to obtain total sensitivity indices by each tool, revealing significant computational gains by applying the proposed methodology. Note that the execution time was more significant for Python than R, indicating that the programming of the Sobol–Jansen method in Python must be improved.

**Table 3.** Execution time for the Sobol–Jansen method using SMs and SAG mill.

| Approach | Power Consumption | | Comminution-Specific Energy | |
|---|---|---|---|---|
| | Python, s | R, s | Python, s | R, s |
| SVR | 104.47 | 1.10 | 104.78 | 0.87 |
| RFR | 34.42 | 2.61 | 35.11 | 2.45 |
| MLP | 25.84 (skl), 140.86 (tens.) | 0.43 | 26.10 (skl), 137.51 (tens.) | 0.40 |
| LR | 25.09 | 0.47 | 24.86 | 0.48 |
| ENR | 24.94 | - | 24.47 | - |
| RR | 24.82 | - | 24.63 | - |
| LAR | 24.7 | - | 24.70 | - |
| GPR | 230.38 | - | 226.85 | - |
| ELM | - | 2.06 | - | 2.22 |
| Grinding model | - | 18,500 | - | - |

### 3.2. Cell Bank: Batch Training

We considered the rougher bank shown in Figure 6, modeled using the expressions proposed by Hu et al. [45]. This model considered the entrained flotation recovery, the true flotation recovery, the sedimentation velocity of the particle, the viscosity of the slurry, and the particle size, among others. For this reason, the flotation model was challenging to solve. Under these conditions, this was coded in the GAMS software and solved using the BARON solver, which is widely used in optimization. The model input variables and their standard values (extracted from [45]) are shown in Table 4.

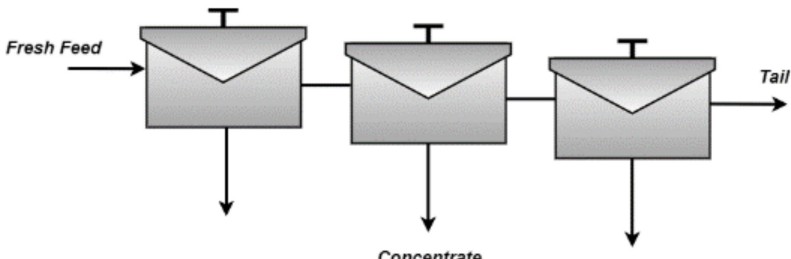

**Figure 6.** Schematic representation of a cell bank.

**Table 4.** Input variables from the cell bank model.

| Label. | Input Variables | Operational Conditions | Unit | Uncertainty |
|---|---|---|---|---|
| $x_1$ | First cell froth depth ($h_{f1}$) | 0.200 | m | $U[0.15, 0.25]$ |
| $x_2$ | Second cell froth depth ($h_{f2}$) | 0.200 | m | $U[0.15, 0.25]$ |
| $x_3$ | Third cell froth depth ($h_{f3}$) | 0.200 | m | $U[0.15, 0.25]$ |
| $x_4$ | Gangue flotation rate constant ($k_g$) | 0.0008 | $h^{-1}$ | $U[6 \times 10^{-4}, 1 \times 10^{-3}]$ |
| $x_5$ | Chalcopyrite flotation rate constant ($k_c$) | 40 | $h^{-1}$ | $U[30, 50]$ |
| $x_6$ | Gangue flowrate ($Q_g$) | 155 | $m^3/h$ | $N[155, 1]$ |
| $x_7$ | Chalcopyrite flowrate ($Q_c$) | 0.3 | $m^3/h$ | $N[0.3, 0.01]$ |
| $x_8$ | First cell superficial air velocity ($v_{g1}$) | 0.007 | m/s | $U[6 \times 10^{-3}, 8 \times 10^{-3}]$ |
| $x_9$ | Second cell superficial air velocity ($v_{g2}$) | 0.007 | m/s | $U[6 \times 10^{-3}, 8 \times 10^{-3}]$ |
| $x_{10}$ | Third cell superficial air velocity ($v_{g3}$) | 0.007 | m/s | $U[6 \times 10^{-3}, 8 \times 10^{-3}]$ |
| $x_{11}$ | First cell interface bubble size ($r_{in1}$) | 0.00045 | m | $U[3 \times 10^{-4}, 6 \times 10^{-4}]$ |
| $x_{12}$ | Second cell interface bubble size ($r_{in2}$) | 0.00045 | m | $U[3 \times 10^{-4}, 6 \times 10^{-4}]$ |
| $x_{13}$ | Third cell interface bubble size ($r_{in3}$) | 0.00045 | m | $U[3 \times 10^{-4}, 6 \times 10^{-4}]$ |
| $x_{14}$ | Particle size ($d_p$) | 55 | μm | $U[30, 80]$ |

The copper recovery and concentrate grade were selected as the output variables of the cell bank. The distribution functions used to describe the uncertainty on the input variables are shown in Table 4. Initially, the flotation model was solved considering some samples of the operational conditions, requiring an average of 10 s to solve one operational instance. The Sobol–Jansen method requires a large size sample and resample to provide reliable results. This way, the execution time of this analysis would ascend approximately to 28,000,000 s, obstructing the analysis of the outcomes. It is worth mentioning that the Sobol-Jansen method is unavailable in GAMS, and its programming in this software implies a challenge. Therefore, as it was seen in the previous application, it was necessary to develop an SM. In Step 1, the LHS sampling method was used to generate the operating conditions of the cell bank. In Step 2, previous samples were used to simulate the cell bank model. In Step 3, the samples with corresponding responses were utilized to develop SMs, which are shown in Figure 7.

Figure 7 shows the results obtained during the SML tool training and testing using 80 and 20% of the dataset, respectively. This figure does not show RFR-r because its performance was very poor; its inclusion was detrimental to achieving a good resolution of the performance of the rest of the SML tools. Again, the SML tool multiparameters were tuned via trial and error. In general, Figure 7 reveals that as the dataset increased from 600 to 2000, R2 increased for both the SML tool training and testing. This increase in samples benefited the capture of the cell bank's behavior regarding uncertainty space, and in turn, the SML tools' performance. This figure shows that SVR-py, MLP-skl, LR-skl, ENR-py, RR-py, LAR-py, GPR-py, MLP-TensorFlow-py, MLP-r, and ELM-DEA-r provided an R2 close to 0.8 when the simulation sample was equal to 2000; then, they went to the next stage. In Step 4, the GSA was carried out using the Sobol–Jansen method and SMs, and the results are shown in Figures 8 and 9.

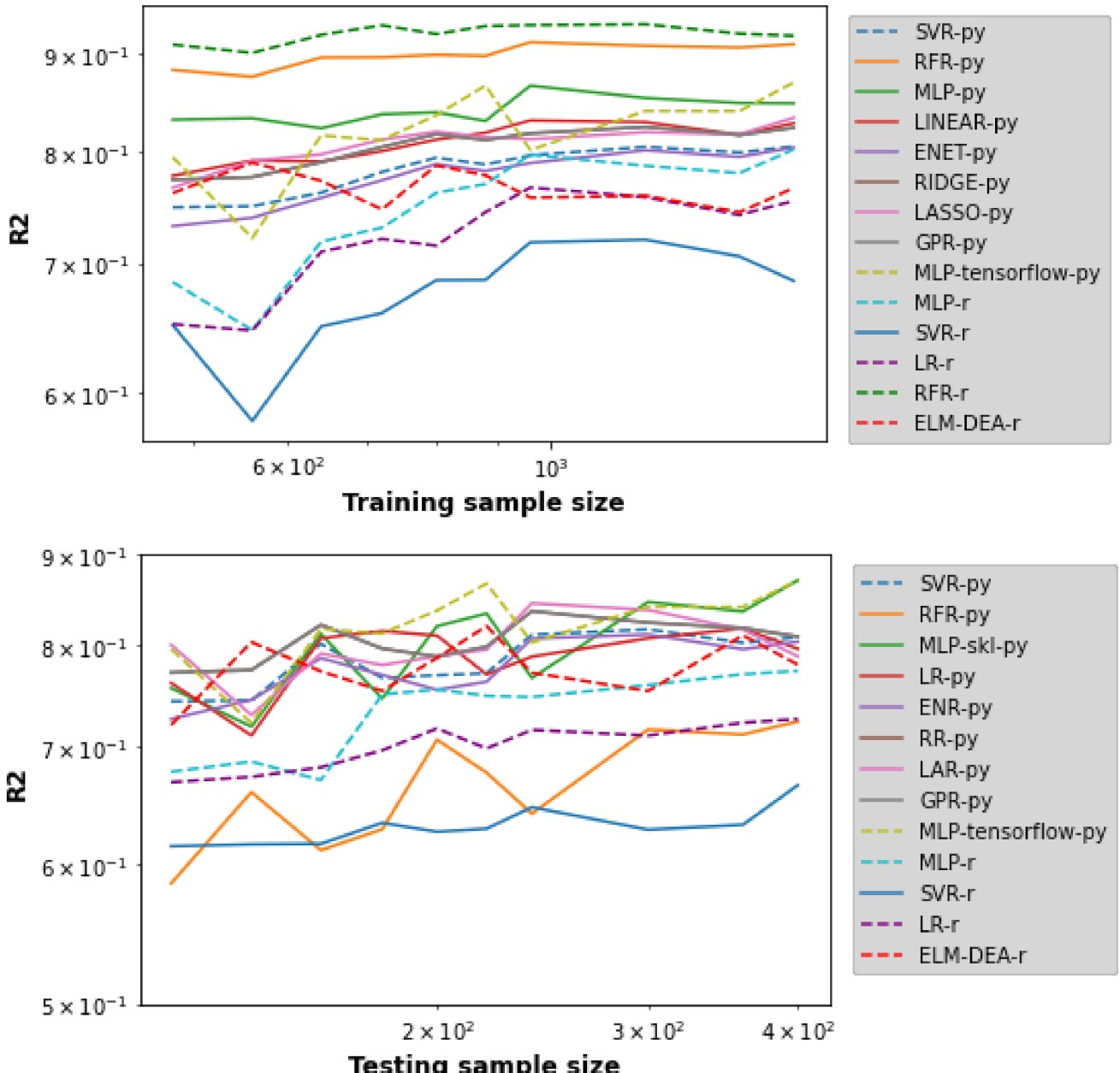

**Figure 7.** Performance of SML tools to construct SMs for cell bank outcomes.

Figure 8 shows the total sensitivity indices provided by the Sobol–Jansen method for the SMs when the output variable was copper grade. Here, we see that the total sensitivity indices provided by GSA were similar. These indicated that $v_{g1}$, $v_{g2}$, $v_{g3}$, and $d_p$ influenced the cell bank's copper grade. The particle size and superficial air velocity influenced the particle–bubble collision efficiency, and consequently, they influenced true and entrained flotation recoveries [45–48]. These results implied that the uncertainty space can be reduced to $(h_{f1} = 0.2$, $h_{f2} = 0.2$, $h_{f3} = 0.2$, $k_g = 8 \times 10^{-4}$, $k_c$, $Q_g = 155$, $Q_c$, $v_{g1}$, $v_{g2}$, $v_{g3}$, $r_{in1}$, $r_{in2}$, $r_{in3}$, $d_p)$ for the copper grade. In other words, the SM concerning copper grade can be reduced from 14-dimensional to 9-dimensional.

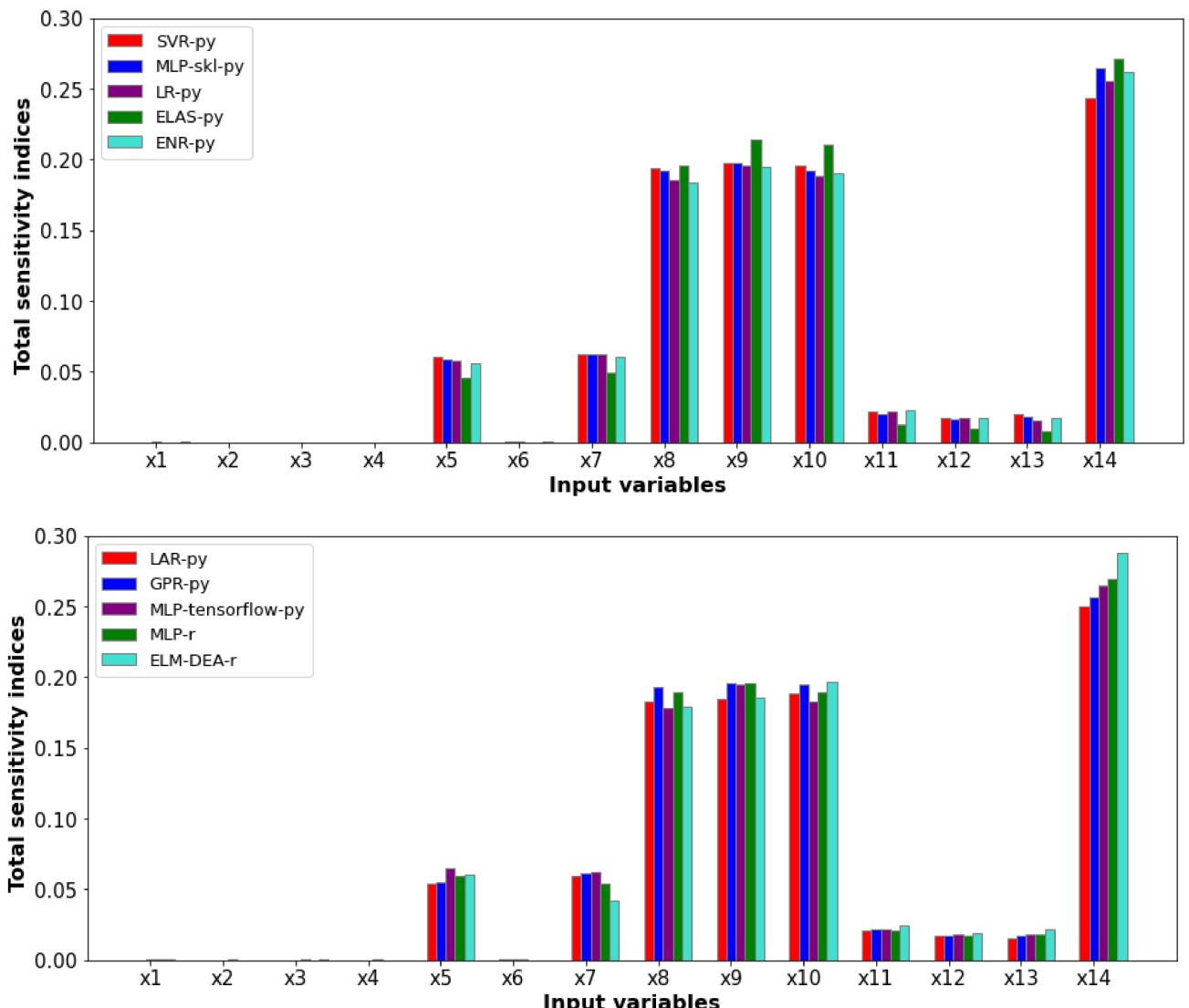

**Figure 8.** Total sensitivity indices obtained via the Sobol–Jansen method and SMs, copper grade.

Figure 9 shows the total sensitivity indices for the SMs when the output variable was copper recovery. These indices were similar and indicated that $k_c$ and $d_p$ influenced the cell bank's copper recovery. The flotation rate constant is directly related to the chalcopyrite mass flow present in the concentrate [45]. Particle size directly influences the contact probability with the bubbles ("elephant curve"), as well as its settling velocity, and consequently, the true and entrainment flotation recoveries [45,47]. Thus, the SM concerning copper recovery can be simplified from 14-dimensional to 5-dimensional. Therefore, the Sobol–Jansen method allowed for simultaneously quantified results reported only from an experimental point of view. Input variables $v_{g1}$, $v_{g2}$, $v_{g3}$, and $d_p$ must be analyzed to optimize the cell bank copper grade, and $k_c$ and $d_p$ must be studied to optimize the copper recovery. Table 5 shows the execution time required to obtain total sensitivity indices by each tool, revealing significant computational benefit by applying the methodology proposed. Again, the execution time was more significant for Python than R, revealing that the programming of the Sobol–Jansen method in Python must be enhanced.

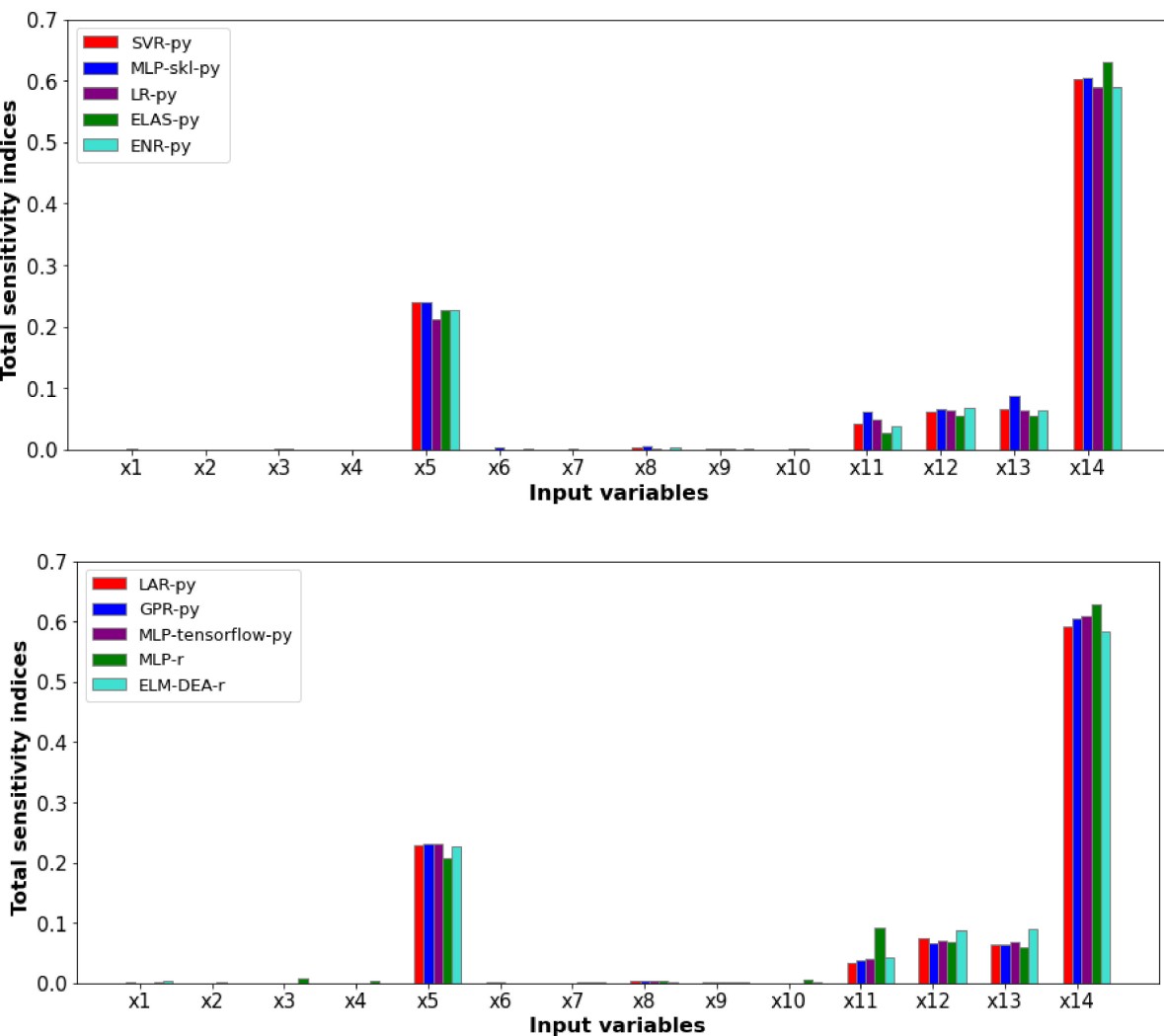

**Figure 9.** Total sensitivity indices obtained via the Sobol–Jansen method and SMs, copper recovery.

**Table 5.** Execution time for the Sobol–Jansen method using SMs, cell bank.

| | Copper Grade | | | Copper Recovery | |
|---|---|---|---|---|---|
| Approach | Python, s | R, s | GAMS, s | Python, s | R, s |
| SVR | 486.31 | 3.44 | - | 496.38 | - |
| RFR | 72.46 | 1.21 | - | 73.22 | - |
| MLP | 54.61 (skl), 283.02 (tens.) | 0.48 | - | 59.07 (skl), 296.95 (tens.) | 0.58 |
| LR | 55.28 | 3.34 | - | 54.54 | - |
| ENR | 55.05 | - | - | 55.44 | - |
| RR | 55.21 | - | - | 55.18 | - |
| LAR | 55.42 | - | - | 55.94 | - |
| GPR | 783.03 | - | - | 761.25 | - |
| ELM | - | 3.39 | - | - | 3.48 |
| Bank model | - | - | 28 M | - | - |

### 3.3. SAG Mill: Online Sequential Training

The EML-DEA surrogate model provided sensitivity indices close to those obtained via the grinding model; however, the training time required makes its application difficult if the dataset arrives sequentially. Under these conditions, the online OS-ELM is presented as a feasible alternative, whose implementation is illustrated as follows. First, we consider that 700 samples arrive for the training of batch ELM; second, 10 sets arrive 30-by-30, which are used to train sequentially via Algorithm 1. The results obtained are shown in Table 6.

**Table 6.** Online sequential training of ELM, SAG mill.

| | | R2-Training | | | |
|---|---|---|---|---|---|
| **Chunk** | **Outputs** | **Initial Dataset** | **New Dataset** | **Total Dataset** | **R2-Testing** |
| 1 | $P_w$ | 0.9792747 | 0.9676392 | 0.9788141 | 0.9714207 |
| | $E_{cs}$ | 0.9872921 | 0.9801317 | 0.9869788 | 0.9862562 |
| 2 | $P_w$ | 0.9792043 | 0.9656083 | 0.9783193 | 0.9713839 |
| | $E_{cs}$ | 0.9872607 | 0.9842905 | 0.9869990 | 0.9862292 |
| 3 | $P_w$ | 0.9793019 | 0.9745175 | 0.9788893 | 0.9715430 |
| | $E_{cs}$ | 0.9859681 | 0.9445565 | 0.9799809 | 0.9853455 |
| 4 | $P_w$ | 0.9793019 | 0.9745605 | 0.9785515 | 0.9717058 |
| | $E_{cs}$ | 0.9854533 | 0.9563790 | 0.9795342 | 0.9850057 |
| 5 | $P_w$ | 0.9792412 | 0.9772665 | 0.9788337 | 0.9717603 |
| | $E_{cs}$ | 0.9846342 | 0.9568821 | 0.9781478 | 0.9843283 |
| 6 | $P_w$ | 0.9792202 | 0.9771648 | 0.9787545 | 0.9717825 |
| | $E_{cs}$ | 0.9838239 | 0.9573903 | 0.9771821 | 0.9836896 |
| 7 | $P_w$ | 0.9792254 | 0.9775622 | 0.9788150 | 0.9717774 |
| | $E_{cs}$ | 0.9831072 | 0.9589930 | 0.9765053 | 0.9831207 |
| 8 | $P_w$ | 0.9791253 | 0.9767157 | 0.9784549 | 0.9717293 |
| | $E_{cs}$ | 0.9822101 | 0.9601473 | 0.9756296 | 0.9823725 |
| 9 | $P_w$ | 0.9790951 | 0.9753149 | 0.9779454 | 0.9717686 |
| | $E_{cs}$ | 0.9809689 | 0.9612125 | 0.9745500 | 0.9812378 |
| 10 | $P_w$ | 0.9790826 | 0.9749347 | 0.9777599 | 0.9717378 |
| | $E_{cs}$ | 0.9800560 | 0.9634199 | 0.9742663 | 0.9803943 |

Table 6 shows that R2 for training and testing was greater than 0.8 in both power consumption and specific comminution energy and for all chunks including the batch ELM (first chunk). This procedure was repeated, considering the arrival of 10 sets 20-by-20. Again, good results were obtained from the point of view of the solution quality and execution time (approximately 0.1 s for each chunk). Each SM developed was immediately subjected to GSA via the Sobol–Jansen method using different magnitudes for uncertainties, as shown in Figure 10.

Figure 10 shows the sensitivity indices provided by the Sobol–Jansen method and SMs sequentially constructed. The GSA considered an uncertainty between 5 and 10% on the input variables' uncertainty magnitude. We can see that the uncertainty magnitude changed the influence of the input variables on the SAG mill's outcomes. For instance, ore hardness uncertainty-magnitude variations generated total sensitivity indices between 0.135 and 0.3 and between 0.08 and 0.17 for power consumption and specific comminution energy, respectively. These results are promising for the application of online GSA for mineral processing. However, depending on the SML tool utilized, the Sobol–Jansen method requires a few seconds to provide the total sensitivity indices; this time could be reduced via the parallelization of the Sobol–Jansen method using packages/libraries such as doParallel [49], Multidplyr, and Numba [50], among others.

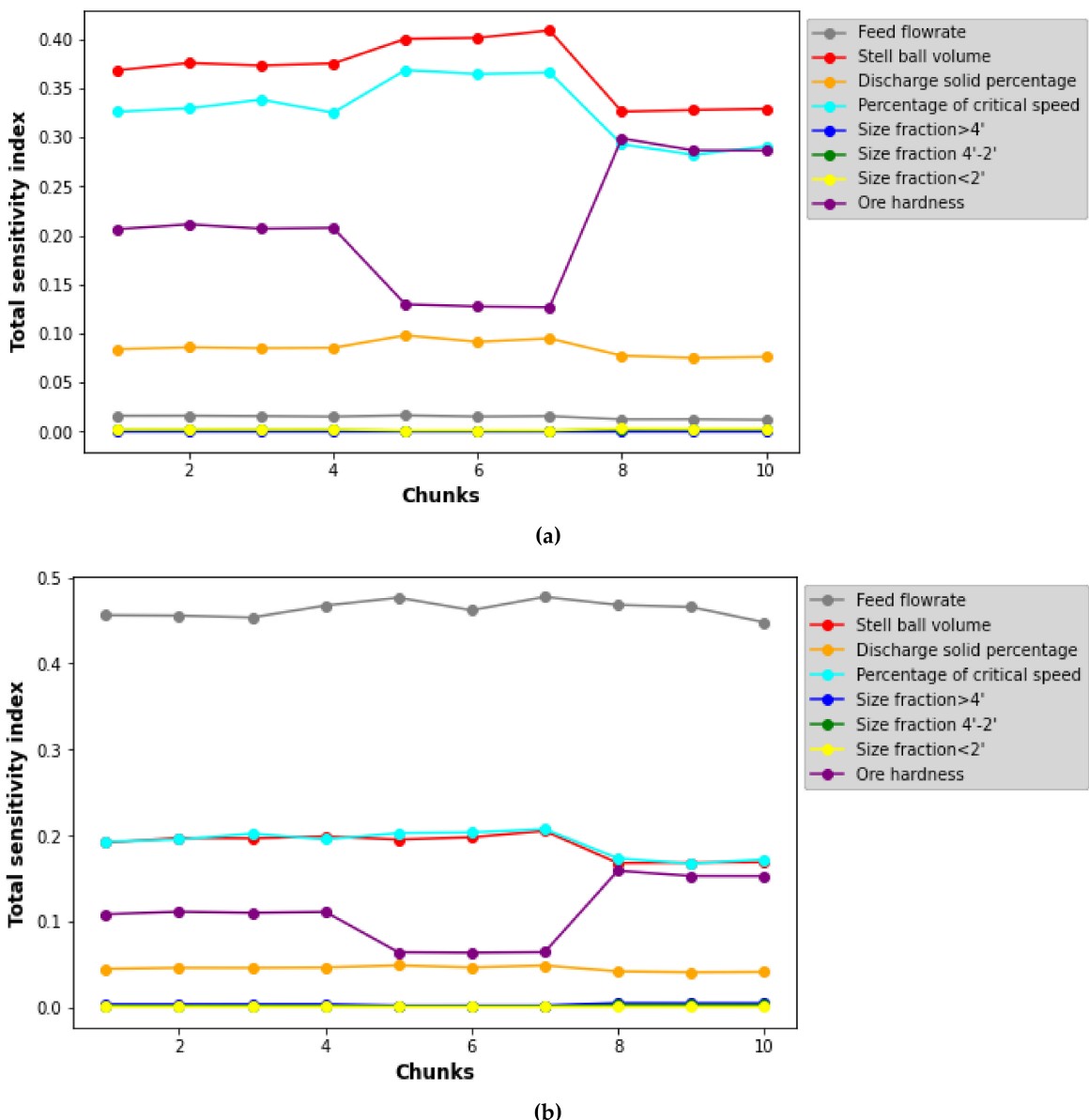

**Figure 10.** Sobol indices with different magnitudes of uncertainty for SMs built via online sequential training. (**a**) Power consumption, (**b**) Specific comminution energy.

## 4. Conclusions

A methodology to build surrogate models for accelerating global sensitivity analysis was presented. This method considers mining processing simulators, experiment design, supervised machine learning tools, and Sobol-based GSA methods. The procedure was illustrated via three case studies, including an SAG mill and a cell bank, batch and online sequential training, and evaluating libraries/packages included in the Python and R programming languages. The main outcomes of our work are the following:

- Surrogate models allow for a reduction in the execution time of GSA from hours or days to seconds, revealing significant computational gains for the methodology proposed.
- In general, the tensorflow and scikit-learn libraries included in Python provide better surrogate models than the packages included in R.
- The effect of critical input variables on metallurgical process performance was quantified, including ore hardness, ore size, and superficial air velocity, among others, which were reported in the literature only from an experimental standpoint.

- The online sequential-ELM provides a good performance regarding solution quality and execution time.
- GSA-OS-ELM opens the door to estimating online sensitivity indices for devices used in mineral processing.

**Supplementary Materials:** The following supporting information can be downloaded at: https://www.mdpi.com/article/10.3390/min12060750/s1, scripts for GSA and surrogate models.

**Author Contributions:** Conceptualization, F.A.L.; methodology, F.A.L.; software, F.A.L.; formal analysis, F.A.L.; investigation, F.A.L.; data curation, F.A.L.; writing—original draft preparation, F.A.L.; writing—review and editing, F.A.L. All authors have read and agreed to the published version of the manuscript.

**Funding:** The authors are grateful for the support of Agencia Nacional de Investigación y Desarrollo de Chile (ANID) through Anillo-Grant No. ACT210027 and Fondecyt 1211498.

**Acknowledgments:** This publication was supported by ANID, Anillo-Grant ACT210027, and Fondecyt 1211498.

**Conflicts of Interest:** The author declares no conflict of interest.

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
