# Peer review of "Accelerating Global Sensitivity Analysis via Supervised Machine Learning Tools: Case Studies for Mineral Processing Models"

_minerals, doi:10.3390/min12060750_

Round 1

Reviewer 1 Report

The paper is an important asset in verifying the potentials of global sensitivity analysis application in mineral processing. Overall, the paper is very well designed and written. I believe that only minor changes are necessary. The suggestions on these are in the attached pdf.

Author Response

We are grateful to the reviewers for their valuable comments. The answers are below, and all introduced changes are highlighted in yellow, green, or grey in the revised version of the manuscript. English editing changes are not highlighted in color, but an editing certificate is attached at the end.

Reviewer 1 (yellow)

1 line 36

R. “Mining processes” was changed by mineral processing

2. line 40

R “mining devices” was changed by mining processing equipment

3. Line 51

R. The following text was added in the revised version of the manuscript “GSA can be applied using different approaches, such as screening, linear regression-based, and variance decomposition-based methods, the latter highlighted for their efficiency and versatility. Within this context, variance decomposition-based methods have been applied in mineral processing, e.g., flotation [1], grinding [2], mineral leaching [3], and lithium ore processing[4].”

4. Line 53

R. The following sentence was added in the revised version of the manuscript “However, these works did not quantify the impact of critical input variables on the metallurgy performance of the processes; for example, the effect of ore hardness on grinding power consumption or the impact of superficial air velocity and particle size on flotation copper grade was not studied”

5. Line 60

R. In the revised version of the manuscript was added the following text “surrogate models (SMs)”

6. Line 74.

R. In the revised version of the manuscript was added the following sentence “Machine learning (ML)”

7. Line 78, 84, 131, and 137.

R. The reference was improved.

8. Line 99

R. The abbreviature SA was added in the sentence.

9. Line 158

R. The author’s name was corrected from “Guang-Bin et al.” to ”Guang et al.”

10 Line 164

R. The sentence “Guang-Bin and their co-workers” was changed by “Guang and co-workers” in the revised version of the manuscript

11 Line 194.

R. A reference was added in the sentence “According to related literature [1]”

12. Line 198

R. The reference was corrected.

13. Line 210

R. “An” was changed by “A” in the revised version of the manuscript.}

14. Line 220

R. “mining model” was changed by “mining processing model”

15. Line 242

R. The point was deleted

16. Line 428

R. The point was deleted.

17.  Line 270

R. The text “Then, this was coded by the authors” was changed by “Within this context, this method was coded by the author”

18. Line 282

R. The sentence “time execution” was modified by “execution time”

19. Line 365.

R. The word “gange” was modified by “gangue”

20. Line 367

R. The word “variable” was changed by “variables”

21. Line 444

R. The bold was checked

22. Line 483

R. The word “processing” was incorporated in the sentence “mining processing simulators”

23. Line 549

R. The reference was checked.

24. Line 560

R. The reference was checked.

25. Line 602

R. The duplicate reference was deleted

Reviewer 2 Report

This is an interesting paper showing the clear merits of using surrogate models and machine learning for conducting global sensitivity analysis. The combination with online sequential training is also promising. The results are interesting and well presented, however in addition to addressing the following concerns by the reviewer, some major text and language editing is still required to improve readability and understanding:

11.       Line 90: “the technique called -as extreme learning machine (ELM) developed by the authors”. The authors mean here that they have developed python packages for ELM, and not developed the ELM as a technique, right?

22.       Line 273: Do the authors mean as a ratio, or 1000 as an absolute number? This should differ based on the number of input parameters investigated and the problem size

33.       Figure 3: Were the same tuning parameters used when the same models were used in Python and in R? why is there a difference in the performance of RFR in python and R? Also why is there a drop in accuracy with RFR-r with the training sample size? this drop is not reflected in the testing set

44.       Line 286: “whose results are shown in Figure 2.“ Figure 2 doesn’t show results, it shows the SAG mill.

55.       Figure 7: why is there no RFR-r in the testing graph?

66.       Figure 7: maintain the same abbreviations in the legend (e.g., LINEAR and LR). Also, it would be better to show the training data first to be consistent with Figure 3 

77.       Line 462: What was proposed by Liang et al. [46]. This phrase doesn’t seem to fit the rest of the sentence. Please clarify or rephrase.

88.       Figure 10: add the (a) and (b) labels to the figure in line with the caption

99.       Line 472: “generate total sensitivity indices between 0.15-0.3”. Figure 10 (a) shows values that are lower than 0.15. Please correct.

110.   Figure quality should be improved 

111.   Generally, the text requires a language review. In addition to what was mentioned above, here are some of the instances, which were difficult to read: Line 9, Line 51, line 70, line 72, line 132

Author Response

I am grateful to the reviewers for their valuable comments. The answers are below, and all introduced changes are highlighted in yellow, green, or grey in the revised version of the manuscript. English editing changes are not highlighted in color, but an editing certificate is attached at the end.

Reviewer 2 (green)

1. Line 90: “the technique called -as extreme learning machine (ELM) developed by the authors”. The authors mean here that they have developed python packages for ELM, and not developed the ELM as a technique, right?

R. ELM was coded for the R kernel aiming to compare different approaches. Specifically, machine learning tools offered by R or Python are trained using exact algorithms (backpropagation[5], sequential minimal optimization [6], Levenberg-Marquardt [7], among others). In contrast, ELM was trained using an approximate algorithm such as differential evolution optimization. The code developed for ELM is standard, i.e., it lacks an executable file for its use by any users. The objective of the manuscript is not to offer a package for the users. Therefore, the following text was added in the manuscript’ revised version, “To determine the best tools to build SMs, benchmarking is confected considering libraries/packages included in Python and R programming languages. These libraries/packages offer SML tools trained via exact algorithms; in order to spice up this benchmarking, the technique known as extreme learning machine (ELM) was coded by the author and trained via an approximate algorithm.”

2. Line 273: Do the authors mean as a ratio, or 1000 as an absolute number? This should differ based on the number of input parameters investigated and the problem size.

R. The following text was added in manuscript’s revised version, “According to Pianosi et al. [8], the sample size commonly used is 500 to 1000 [9]. However, the researcher indicated that the sample size may vary significantly from one application to another. Therefore, a much larger sample might be needed to reach reliable results. Additionally, the number of samples required to achieve stable sensitivity indices can vary from one input variable to another, with low sensitivity inputs usually converging faster than high sensitivity ones [10]. In this first instance, a sample size equal to 1000 allows achieving stable sensitivity indices. Therefore, the sample size used for carrying out GSA was 16,000 data.”

3. Figure 3: Were the same tuning parameters used when the same models were used in Python and in R? why is there a difference in the performance of RFR in python and R? Also why is there a drop in accuracy with RFR-r with the training sample size? this drop is not reflected in the testing set.

R. The following text was added in the revised version of the manuscript, ”Before they proceeded to the next step, note that RFR-r and RFR-py exhibited disparate performances. RFR, offered by the RandomForest package (RFR-r) and the scikit-learn library (RFR-p), is based on Breiman’s original version [11] and Geurts et al. [12], respectively. The latter consists of randomizing strongly both input variables and cut-point while splitting a tree node. In the extreme case, this approach randomly chooses a single input variable and cut-point at each node, building totally randomized trees whose structures are independent of the output variable values of the learning sample. Therefore, RFR versions offered by scikit-learn and RandomForest differ, which could explain the disparity in their performance. In the case of RFR-r, its decreasing performance could be related to perturbations induced in the models by modifying the algorithm responsible for the search of the optimal split during tree growing [12]. In this context, the researcher indicated that it is productive to deteriorate somewhat the algorithm’s “optimality” that seeks the best split locally at each tree node.”

4. Line 286: “whose results are shown in Figure 2.“ Figure 2 doesn’t show results, it shows the SAG mill.

R. In the revised version of the manuscript, Figure 2 was changed by Figure 3.

5.   Figure 7: why is there no RFR-r in the testing graph?

R. The following text was added in the revised version of the manuscript, “This figure does not show RFR-r because its performance was very poor; its inclusion was detrimental to achieving a good resolution of the performance of the rest of the SML tools”

6. Figure 7: maintain the same abbreviations in the legend (e.g., LINEAR and LR). Also, it would be better to show the training data first to be consistent with Figure 3

R. The legend “LINEAR” was changed by “LR”. Besides, in the revised version, training results are shown first than testing results.

7. Line 462: What was proposed by Liang et al. [46]. This phrase doesn’t seem to fit the rest of the sentence. Please clarify or rephrase.

R. In the revised version of the manuscript, the sentence named was deleted

8. Figure 10: add the (a) and (b) labels to the figure in line with the caption

R. In the revised version of the manuscript, (a) and (b) labels were added in Figure 10

9. Line 472: “generate total sensitivity indices between 0.15-0.3”. Figure 10 (a) shows values that are lower than 0.15. Please correct.

R. You are right; in the revised version of the manuscript, the range 0.15-0.3 was changed by 0.13-0.3.

10. Figure quality should be improved 

R. The quality of Figure 10 was improved.

11. Generally, the text requires a language review. In addition to what was mentioned above, here are some of the instances, which were difficult to read: Line 9, Line 51, line 70, line 72, line 132.

R. The manuscript was sent to English edition; the certificate is attached.

Reviewer 3 Report

The study presents a novel idea to improve the efficacy of metal ore mining by using artificial intelligence. The authors built a neural network and tested the output variations by using the global sensitivity methodology. The text is clearly written and contains all the needed information. Besides, it is very good that both python and r were tested with the same models. Only a few remarks can be found. 

  • There are a few typos to correct. Please, correct the English in Fig. 1. Consider deleting the abbreviations that are not regularly and intensively used throughout the text.
  • A model verification check would be convenient, but not necessary in this case. 
  • Furthermore, r2 is not a sufficient statistical test to make the model acceptable.

Author Response

I am grateful to the reviewers for their valuable comments. The answers are below, and all introduced changes are highlighted in yellow, green, or grey in the revised version of the manuscript. English editing changes are not highlighted in color, but an editing certificate is attached at the end.

Reviewer 3 (grey)

The study presents a novel idea to improve the efficacy of metal ore mining by using artificial intelligence. The authors built a neural network and tested the output variations by using the global sensitivity methodology. The text is clearly written and contains all the needed information. Besides, it is very good that both python and r were tested with the same models. Only a few remarks can be found. 

1. There are a few typos to correct. Please, correct the English in Fig. 1. Consider deleting the abbreviations that are not regularly and intensively used throughout the text.

R. The manuscript was sent to English editing; the certificate is attached.

2. A model verification check would be convenient, but not necessary in this case. 

R. The grinding model was validated in work [13]. The bank model results are validated by reviewing the literature. For instance, the impact of the particle size on recovery and copper grade has been reported by several authors from an experimental point of view. The GSA allows quantifying the impact of particle size on metallurgical parameters from a simulation point of view.

3. Furthermore, r2 is not a sufficient statistical test to make the model acceptable.

R. The following text was added in the manuscript’ revised version, “It is worth mentioning that there are several measures to evaluate the performance of surrogate models. Prominent among these are R2, mean squared error (MSE), root mean squared error (RMSE), and mean absolute error (MAE) [14]. The latter three can vary from 0 to any larger number, whereas R2 exhibits values between 0 and 1. An R2 value closer to 1 means that the regression model covers most parts of the variance of the values of the response variable and can be termed as a good model. In contrast, with MSE, RMSE, and MAE values depending on the scale of values of the response variable, the value will be different and, hence, it is difficult to assess for certain whether the regression model is good or otherwise.” Therefore, in the manuscript, r2 was used to evaluate the SMs’ performance.”
